# Evolving Trends and Patterns of Utilization of Magnetic Resonance-Guided Radiotherapy at a Single Institution, 2018–2024

**DOI:** 10.3390/cancers17020208

**Published:** 2025-01-10

**Authors:** Robert A. Herrera, Eyub Y. Akdemir, Rupesh Kotecha, Kathryn E. Mittauer, Matthew D. Hall, Adeel Kaiser, Nema Bassiri-Gharb, Noah S. Kalman, Yonatan Weiss, Tino Romaguera, Diane Alvarez, Sreenija Yarlagadda, Ranjini Tolakanahalli, Alonso N. Gutierrez, Minesh P. Mehta, Michael D. Chuong

**Affiliations:** 1Department of Radiation Oncology, Miami Cancer Institute, Miami, FL 33176, USA; eyub.akdemir@baptisthealth.net (E.Y.A.); rupeshk@baptisthealth.net (R.K.); kathrynm@baptisthealth.net (K.E.M.); matthewha@baptisthealth.net (M.D.H.); nema.bassirigharb@baptisthealth.net (N.B.-G.); noahk@baptisthealth.net (N.S.K.); yonatan.weiss@baptisthealth.net (Y.W.); antinogenesr@baptisthealth.net (T.R.); dianeal@baptisthealth.net (D.A.); sreenija.yarlagadda@baptisthealth.net (S.Y.); ranjinit@baptisthealth.net (R.T.); alonsog@baptisthealth.net (A.N.G.); mineshm@baptisthealth.net (M.P.M.); 2Herbert Wertheim College of Medicine, Florida International University, Miami, FL 33199, USA

**Keywords:** MR-guided radiotherapy, MRgRT, stereotactic ablative radiotherapy, on-table adaptive radiation therapy, stereotactic body radiotherapy, SBRT, SMART, ART, ultra-hypofractionated, treatment patterns

## Abstract

Magnetic resonance-guided radiotherapy (MRgRT) is expanding worldwide thanks to advances in soft tissue imaging, continuous visualization of the target and normal organs-at-risk during treatment, automated intelligently gated beam delivery within predefined targeting boundaries, and on-table adaptive replanning, all of which permit improved treatment efficacy, toxicity reduction, and shortened fractionation regimens. This, however, is still a nascent technology which can be more time- and resource-intensive than standard radiotherapy, and hence its optimal utilization and deployment remain in constant flux and evolution. We retrospectively analyzed our institutional MRgRT utilization across 823 treatment courses over a 6-year period, which predominantly included abdominal and pelvic tumors treated with dose-escalated ultra-hypofractionation.

## 1. Introduction

Contemporary image-guided radiotherapy (IGRT) relies on kilovoltage/megavoltage (kV/MV) portal imaging and/or cone-beam computed tomography (MV or kvCT)-based techniques to ensure appropriate patient positioning and target localization. However, these imaging modalities may provide a suboptimal visualization of gross disease and organs-at-risk (OARs) due to limitations in visualizing low-density structures, especially when adjacent or abutting [1]. Magnetic resonance (MR)-guided radiotherapy (MRgRT) is a novel technology featuring advanced imaging and rapid replanning capabilities using on-table images [2] that may improve clinical outcomes by facilitating safe dose escalation and reducing toxicity, especially for tumors in challenging anatomic locations [3,4] that have suboptimal outcomes when treated with CT-guided linear accelerators (Linacs) [3,5,6,7,8,9].

In 2018, our institution became one of the first worldwide adopters of a 0.35-Tesla (T) MR-Linac (ViewRay Inc., Cleveland, OH, USA) [10]. The 0.35-T MR-Linac has several advanced capabilities including continuous intrafraction multi-planar MR imaging [2], automatic beam gating [11], and the ability to deliver on-table adaptive radiotherapy (oART) [6,7]. There has been increasing worldwide MRgRT adoption, and different centers have deployed this technology with differing clinical goal: some focusing on prostate and breast tumors and others focusing on mobile intrathoracic and intrabdominal tumors [12,13,14,15]. Our center has multiple technology platforms, and the goal of incorporating this technology was to implement ablative dosing approaches for mobile tumors, especially susceptible to respiratory incursions, and not suitable for CT-guided approaches because of soft-tissue resolution limitations. Inherently, this focused the indications toward UHfx and oART. These approaches evolved with technology and software improvements as well as with process-based efficiencies and improved learning and QA. We therefore evaluated changes in MRgRT utilization at our institution over a 6-year period with a focus on identifying specific clinical scenarios that might especially benefit from MRgRT, and we also sought to understand whether throughput efficiencies could be achieved with experience.

## 2. Materials and Methods

This single-institution retrospective analysis evaluated patients treated with MRgRT on a 0.35-T MRIdian-Linac between April 2018 and April 2024. Patient demographics, tumor, and treatment data as well as treatment time distributions (e.g., total in-room time [TIRT] and total delivery time [TDT]) were collected (Table 1). We defined TIRT as the time spent inside the treatment room, while TDT was the time from first beam-on to treatment completion. To evaluate trends over time, treatment courses were divided into six consecutive 12-month periods. Patients receiving multiple MRgRT treatment courses for local recurrences or distant tumor sites were treated as distinct cases.

Because we expected a steep learning curve for efficiently and safely treating on the MR-Linac, we began our MRgRT program in April 2018 with a plan to not treat with oART for several months. As such, we delayed implementing oART until September 2018 [6]. A treatment course was classified as oART if at least one fraction (fx) required on-table replanning.

In July 2022, our MR-Linac was upgraded with the A3i system (A3i, 510K approval December 2021) that featured enhanced automation, real-time 3D multi-planar tracking, the BrainTx™ package [16], and a parallel adaptive workflow [17]. Our daily oART workflow is detailed in Figure 1, and Figure 2 provides an example showing the importance of oART for a patient receiving an ablative dose to a mesenteric lymph node. Daily changes in the patient’s position of the small and large bowel relative to the GTV caused dose constraint violations, reducing the target dose coverage and requiring each fraction to be replanned.

Treatment prescription schedules were recalculated and expressed as biological effective dose for tumors, using an α/β of 10 (BED10) [18]. We defined ultra-hypofractionation (UHfx) as courses with five (5) or fewer fxs with a minimum fx size of 5 Gy. Target volumes included a gross tumor volume (GTV), a planning target volume (PTV) with a typical 3 mm expansion (up to 5 mm), and based on specific clinical considerations, a clinical target volume (CTV). To evaluate the anatomic relationship between treated lesions and nearby dose-limiting OARs (e.g., stomach, bowel), we expanded the GTV isotropically by 3 mm and 5 mm. The overlap volumes between the expanded GTV and adjacent OARs was analyzed to estimate areas at risk of underdosing, serving as a surrogate for planning difficulty. Treatment courses were categorized into six 12-month periods, starting from Period 1 (April 2018–April 2019) to Period 6 (April 2023–April 2024).

Patient, disease, and treatment characteristics were summarized using descriptive statistics. Statistical analyses were performed using Microsoft Excel 2022 (Microsoft Corporation, Redmond, WA, USA) and SPSS version 27.0 (IBM, SPSS Statistics, Chicago, IL, USA) with comparisons performed using the Mann–Whitney U test. *p* < 0.05 was considered as statistically significant.

## 3. Results

### 3.1. Patient Cohort

A total of 854 lesions were treated across 823 treatment courses in 712 patients with 6038 delivered fractions. The median age was 69 years (range, 7–94) with the majority being male (444 [53.9%]), White (724 [88.0%]), and Hispanic (452 [54.9%]). Overall, 473 patients (57.5%) underwent treatment to the primary tumor, while 325 (39.5%) received treatment for oligometastatic and 25 (3.0%) for polymetastatic lesions. A total of 40 (4.9%) courses were delivered for re-irradiation, and 18 (2.2%) were delivered as a dose-escalated boost following treatment on a different treatment platform, either proton or photon.

The most frequent sites treated were the pancreas (242 [29.4%]), thorax (172 [20.9%]), abdominopelvic lymph nodes (107 [13.0%]), liver (72 [8.7%]), and adrenal glands (68 [8.3%]). Other (146 [17.7%]) tumor sites included the esophagus (18 [2.2%]), stomach (14 [1.7%]), colorectal (12 [1.5%]), ampulla of vater/bile duct (11 [1.3%]), bone—non-spine (11 [1.3%]), abdominal wall (8 [1.0%]), bone—spine (8 [1.0%]), brain (7 [0.9%]), head and neck (7 [0.9%]), kidneys (7 [0.9%]), mesentery/omentum (5 [0.6%]), vagina (5 [0.6%]), bladder (4 [0.5%]), gallbladder (4 [0.5%]), retroperitoneum (4 [0.5%]), breast (2 [0.2%]), celiac plexus (2 [0.2%]), cervix (2 [0.2%]), paraspinal (2 [0.2%]), pelvic mass (2 [0.2%]), supraclavicular (2 [0.2%]), uterus (2 [0.2%]), axilla (1 [0.1%]), cardiophrenic lymph node (1 [0.1%]), endometrium (1 [0.1%]), inferior vena cave tumor thrombus (1 [0.1%]), ovaries (1 [0.1%]), porta hepatis (1 [0.1%]), and pulmonary artery (1 [0.1%]). The baseline characteristics of patients and tumors are summarized in Table 1.

The number of treatment courses delivered modestly increased over time: 101 in Period 1, 126 in Period 2, 144 in Period 3, 157 in Period 4, 162 in Period 5, and 133 in Period 6. Throughout all periods, the distribution of treated sites remained consistent (Figure 3).

### 3.2. Trends in MR-Guided Radiotherapy

The median prescription dose was 50 Gy (range, 16–76) in a median of 5 fxs (range, 1–41) delivered over a median of 8 days (range, 1–67). The median prescribed BED10 was 94.4 Gy (range, 28.0–200.0). Almost half of all patients (404, 49.1%) were prescribed a highly ablative BED10 ≥ 100 Gy, while almost one third (249, 30.3%) received a BED10 ≥ 70 Gy. Most patients were treated daily (653 [79.3%]) and in breath-hold (687 [83.5%]). An abdominal compression belt (54 [6.6%]) was used for patients who did not tolerate treatment in breath-hold. The median target volumes were 18.7 cc (range, 0.4–1062.0) for GTV, 64.9 cc (range, 1.5–1450.6) for CTV, and 45.7 cc (range, 2.4–2296.5) for PTV.

A total of 6036 fxs were delivered, with 2643 (43.8%) fxs requiring on-table adaptive replanning. The distribution of fxs by treated site was 1353 (22.4%) for pancreas, 1460 (24.2%) for thorax, 631 (10.5%) for abdominopelvic lymph nodes, 307 (5.1%) for adrenal glands, 396 (6.6%) for liver, 91 (1.5%) for prostate, and 1798 (29.8%) for other. Among the 2643 fxs that underwent oART, the distribution was 1081 (17.9%) for pancreas, 484 (8.0%) for thorax, 423 (7.0%) in abdominopelvic lymph nodes, 235 (3.9%) for adrenal glands, 110 (1.8%) for liver, and 310 (5.1%) for other.

Although the vast majority of all courses (778, 94.6%) were multi-fractionated regimens prescribed to a median dose of 50 Gy (range, 16–76; median [range] BED10: 91.7 [18.8–132.0] Gy), a small proportion, (45, 5.5%) of courses were treated with a median single-fx ablative dose of 30 Gy (range, 16–40; median [range] BED10: 120.0 [41.6–200.0] Gy). Among the 45 single-fx courses, the treated sites included thorax (16, 35.6%), liver (9, 20.0%), adrenal glands (7, 15.6%), pancreas (6, 13.3%), abdominopelvic lymph nodes (5, 11.1%), bone—spine (1, 2.2%), and celiac plexus (1, 2.2%).

The median oART prescription was 50 Gy (18–68) in 5 fx (1–36) over 7 days (1–56) in 515 (62.6%) courses. Most treatment courses were delivered daily (427 [51.9%]) in breath-hold (464 [56.4%]). Among the 515 courses, the median BED10 was 100.0 Gy (range, 36.0–157.5) with 277 patients (53.8%) treated with a BED10 ≥ 100 Gy, and an additional 171 (33.2%) were ≥70 Gy. The most commonly treated courses with oART were pancreas (219 [26.6%]), abdominopelvic lymph nodes (91 [11.1%]), adrenal glands (58 [7.0%]), thorax (62 [7.3%]), liver (28 [3.4%]), and other (65 [7.9%]).

One significant pattern of interperiodic temporal evolution was a significant increase in oART, increasing from 168 fxs (19.0%) in Period 1 to 615 fxs (78.9%) in Period 6 (Figure 3). Notably, the percentage of fxs treated with oART for thorax increased from 0.7% to 21.2%, while for pancreas, it rose from 12.7% to 29.0% between Periods 1 and 6. A second significant interperiodic evolutionary change between Periods 1 and 6 was the increase in the use of UHfx courses from 66.3% to 89.5%. Treatment courses that utilized both UHfx and oART increased from 29.7% in Period 1 to 76.7% in Period 6.

The number of courses using oART also substantially increased from 37.6% in Period 1 to 85.0% in Period 6. oART courses nearly doubled from 44.7% to 77.2% from Periods 1–3 to Periods 4–6. When comparing the use of oART across the overall treated sites by period, the use of oART for pancreatic courses had increased from 25.0% in Period 1 to 33.6% in Period 6, while the rate for thoracic tumors had risen from 1.0% in Period 1 to 16.1% in Period 6.

Overlapping dose-limiting OARs within a 3 mm GTV expansion were seen in 58.8% (484) of courses, increasing to 65.9% (542) when a 5 mm GTV expansion was applied. In oART courses, 42.9% (353) of courses had overlap with dose-limiting OARs within the 3 mm GTV expansion, while 48.1% (396) showed overlap within the 5 mm GTV expansion. Pancreatic courses showed the highest overlap with 24.2% (199) of courses showing overlap within the 3 mm GTV expansion and 26.2% (216) within the 5 mm GTV expansion.

The median TDT and TIRT were 15.0 min (IQR, 12.0–20.0) and 51.0 min (IQR, 39.0–64.0), respectively. TIRT was ≤60 min for 573 courses (69.6%) with 408 (49.6%) completed in ≤50 min, 228 (27.7%%) in ≤40 min, and 63 (7.7%) in ≤30 min. The median TDT for free-breathing was 12.0 min (IQR, 9.0–15.0) vs. 16.0 min (IQR, 13.0–21.0) for breath-hold (*p* < 0.001). Similarly, the median TIRT for free-breathing was 43.0 min (IQR, 33.0–55.0) vs. 52.0 min (IQR, 41.0–65.0) for breath-hold (*p* < 0.001). Table 1 presents the TDT and TIRT based on tumor site.

The median TDT for non-adaptive RT and oART was 14.0 min (IQR, 10.0–19.0) and 16.0 min (IQR, 13.0–20.0), respectively (*p* < 0.001), while the median TIRT was 40.0 min (IQR, 32.0–50.0) for non-adaptive RT and 57.0 min (IQR, 47.0–69.0) for oART (*p* < 0.001). Over time, significant improvements were seen in oART with a median TIRT, decreasing from 81.0 min in Period 1 to 45.0 min in Period 6, while TDT decreased slightly from 18.5 min to 16.0 min. Figure 4 presents the TIRT of oART across the six-year period.

## 4. Discussion

We treat a high volume of patients with definitive dose-escalated RT across multiple stereotactic body radiation therapy (SBRT) platforms (TrueBeam [Varian Medical Systems Inc., Palo Alto, CA, USA], CyberKnife [Accuray Inc., Sunnyvale, CA, USA], Tomotherapy [Accuray Inc., Sunnyvale, CA, USA], MRIdian Linac) as well as pencil beam scanning proton therapy. These technologies are all located in the same building, and they are not restricted for use by only a subset of physicians, which facilitates our ability to use a robust peer review process to actively triage patients from the entire practice to the treatment platform that we expect will provide the best therapeutic ratio.

We became an early adopter of the MRIdian Linac in early 2018, and our initial treatment strategy was to attempt significant dose escalation using UHfx to tumors in challenging anatomic locations that we could not safely dose escalate in the same manner using our other treatment devices either because of suboptimal X-ray or CT image guidance and/or the inability to either effectively manage motion or offer oART. Our initial intent was to primarily treat tumors in the abdomen and pelvis based on emerging dose-escalated outcomes from MRIdian cobalt centers demonstrating both safety and encouraging efficacy and also because tumors in these locations are most significantly limited by motion and soft-tissue resolution issues not easily addressed by other technologies [7,19].

While most institutions do not yet offer the MR-Linac platform, there is growing interest in MRgRT that likely has been spurred by increasing clinical evidence demonstrating improved outcomes over conventional forms of RT, specifically in terms of improved local control using high ablative dosing.

In 2022, patterns of utilization analysis from 16 U.S. MRIdian centers indicated that MRgRT was predominantly delivered with UHfx (70.3%) and while oART (38.5%) was relatively uncommon with an average of 1.7 adapted fractions/course, this had grown by a compounded annual rate of 88.5% by the end of 2020 [12]. The most frequently treated sites in the U.S. were the pancreas (20.7%), liver (16.5%), prostate (12.5%), breast (11.5%), lung (9.4%), and “other” organs (10.4%). Similarly, drawing on reports from 21 centers in Europe and Asia, delivering over 46,000 fxs, UHfx schedules constituted 63.5% of courses, with 57.8% requiring oART. The most commonly treated sites were the prostate (23.5%), liver (14.5%), lung (12.3%), pancreas (11.2%), and breast (8.0%) [20].

Since we have been treating with MRgRT for more than half a decade, and because we had specifically focused on minimizing utilization for prostate cancer, the most common use-case as demonstrated in the global survey, our experiential evolution would provide valuable use-case lessons to the radiation oncology community in understanding the value and changing patterns of MRgRT utilization, even in the setting of having access to almost every other major advanced radiotherapeutic platform, thereby specifically identifying patients deriving the greatest benefit from MRgRT. As shown in Figure 2, MRgRT is especially beneficial for patients with highly mobile tumors, particularly in the abdominal and pelvic regions. It allows real-time adjustments for both inter- and intra-fractional motion, improving target volume coverage and positioning to enhance dosimetric parameters. These tumors are highly sensitive to respiratory motion and are often unsuitable for MV or kV CT-guided methods, which are limited by poor soft-tissue resolution and primarily address only inter-fractional motion.

Our MRgRT experience predominantly includes unresectable abdominal tumors that were routinely treated with dose escalation, UHfx, and oART. Abdominal tumors comprised nearly two thirds of all treatment courses in our 6-year experience, predominantly those in the pancreas, and also frequently involving abdominal lymph nodes, liver, and adrenal glands. We published favorable clinical outcomes from our early experience using this approach for various challenging clinical scenarios (e.g., 50 Gy/5 fx for metastatic mesenteric nodules refractory to systemic therapy) [21], and this may have contributed to our steadily increasing volumes across each treatment period that increasingly included patients seeking out MRgRT from outside of south Florida. Our increasing volume, specifically of locally advanced pancreatic cancer, was also likely influenced by favorable outcomes from a recently published phase 2 trial that evaluated the feasibility of ablative 5-fraction stereotactic magnetic resonance-guided on-table adaptive radiation therapy (SMART) for borderline resectable and locally advanced pancreatic cancer; no acute grade ≥ 3 gastrointestinal toxicities definitely related to MRgRT were reported with a very favorable 2-year OS of >50% from diagnosis [22,23]. This aligns with the growing evidence supporting RT as a key component in pancreatic cancer treatment [19,22,24]. Because of the encouraging outcomes that we observed in treating challenging abdominal and pelvic tumors, we increasingly triaged thoracic patients who may specifically benefit with the use of MRgRT, such as central and ultra-central tumors that are at risk for toxicity with non-dose-reduced CT-guided SBRT [25,26]. Additionally, we also frequently utilized the MRIdian Linac for treating peripheral lung tumors with SBRT, especially for those patients either with very poor pulmonary function (such as idiopathic pulmonary fibrosis) or those with peridiaphragmatic lesions subject to 4D-CT verified significant motion. The ability to treat these patients with automatic beam gating, in breath-hold, eliminated the need for an internal target volume (ITV), resulting in significant reduction in normal lung volume irradiated and thus reducing the risk of toxicity.

While UHfx was common in our initial MRgRT experience, the COVID-19 pandemic sharply increased UHfx within our MRgRT program [27,28]. Prior to the pandemic (Period 2), 65.9% of courses were UHfx, rising to 76.4% during the pandemic and reaching almost 90% of all courses in Period 6. The COVID-19 pandemic also led us to consider single-fraction SBRT for the first time [29], which has been increasingly utilized in our practice most commonly for lung and liver metastases. We recently completed the multi-center phase 2 SMART ONE trial of single-fraction SBRT using the MRIdian Linac for tumors in the chest, abdomen, and pelvis, and our initial publication reports feasibility, safety, and efficacy [30]. Since the pandemic, the use of single-fraction SBRT has progressively increased from 5 courses in Period 3 to 12 in Period 4, 20 in Period 5, and 8 in Period 6.

Treatment times on an MR-Linac are typically longer than on a standard linac, which is in large part because delivery uses a step-and-shoot approach but also because all treatments typically incorporate automatic beam gating on the MRIdian Linac and also because the dose-per-fraction is typically very high and the need for precision delivery is significantly greater [31,32]. Reducing treatment times and increasing machine throughput has been a priority for MRgRT centers to improve tolerability for patients—especially those treated with oART (Figure 1). In 2017, Henke et al. reported a median oART TIRT of 79 min for abdominal malignancies treated with a MRIdian cobalt device [7], while Tetar et al. demonstrated an average oART TIRT of 44.7 min for 140 prostate cancer treatments [33]. In 2020, Gungor et al. analyzed 166 oART treatment courses using MRIdian, reporting a median TIRT of 45 min [34]. Prior to our A3i upgrade that introduced several advanced capabilities including a parallel oART workflow allowing physician, physicist/dosimetrist, and radiation therapy to work in parallel on different monitors, we observed a median TIRT of 65 min (IQR, 57–75) across 286 oART courses compared to 47 min (IQR, 40–54) afterwards: a relative time savings of 27.7%. As expected, non-oART treatment times remained stable. When considering all patients treated since 2018, we observed decreasing TIRT that likely was related to technological improvements and also attributable to the development of standardized treatment workflows related to oART as well as increasing experience among all MRgRT team members [35].

There are several limitations of this study including its retrospective design and that our MRgRT utilization is very specific and may not be generalizable to others. For example, we do not treat a high volume of prostate cancer with MRgRT due to having robust referral patterns for abdominal tumors that cannot be safely treated with dose escalation and UHfx on our other treatment machines, and because we have several other very effective prostate cancer treatment platforms. For context, MRgRT has recently been demonstrated to reduce toxicity over CT-guided RT for prostate cancer patients in the randomized MIRAGE trial [3]. Another limitation is our use of a uniform BED α/β ratio of 10, which may not fully account for histology-dependent variations. Lastly, ViewRay, Inc. filed for Chapter 11 bankruptcy in 2023 that resulted in a more limited use of our MR-Linac because of concerns regarding machine downtime and lack of device servicing for a period of time; with recent corporate restructuring, some of these concerns have eased to a certain extent.

## 5. Conclusions

In conclusion, since the inception of our MRgRT program in 2018, we have intentionally prioritized treatment on the MRIdian Linac for patients who would benefit from significant dose escalation above what can be safely delivered on our other advanced radiation treatment systems. The unique capabilities of the MRIdian Linac enable not only safe dose escalation that frequently requires oART, especially for abdominal and pelvic tumors, but also the routine use of UHfx and increasingly single-fraction SBRT. Important reasons for our program’s success include our early decision to develop a standardization of patient selection and oART workflows as well as robust training and credentialing of all staff who are involved in MRgRT within our department.

## Figures and Tables

**Figure 1 cancers-17-00208-f001:**
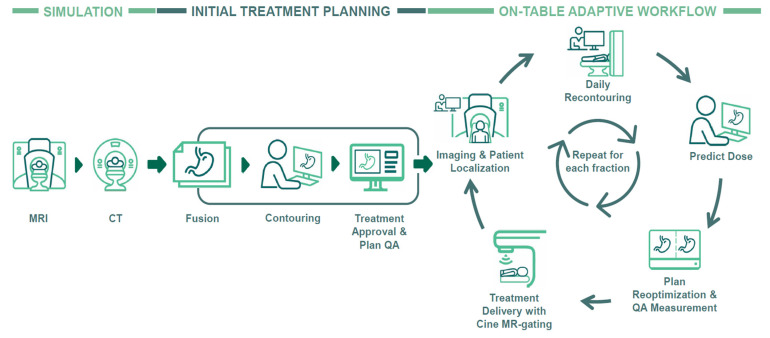
Clinical workflow of magnetic resonance-guided radiotherapy employing on-table adaptive processes. Abbreviations: MRI, magnetic resonance imaging; CT, computer tomography; QA, quality assurance.

**Figure 2 cancers-17-00208-f002:**
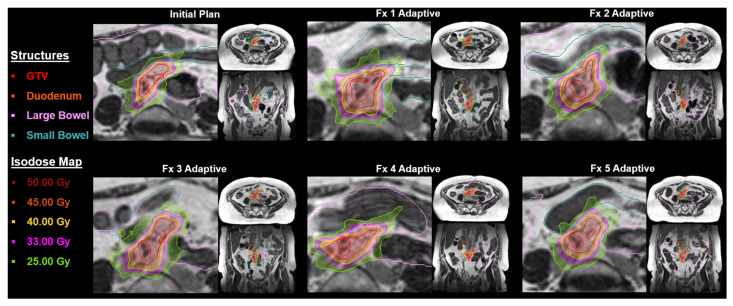
Representation of an on-table adaptive radiotherapy plan for a mesenteric lymph node treated to 50 Gy in 5 fractions. Note the GTV had significant inter-fractional anatomical shifts, necessitating plan adaptation to ensure the preservation of adjacent critical structures and avoidance of any potential violation of organs-at-risk. Abbreviations: GTV, gross tumor volume, Fx, fraction.

**Figure 3 cancers-17-00208-f003:**
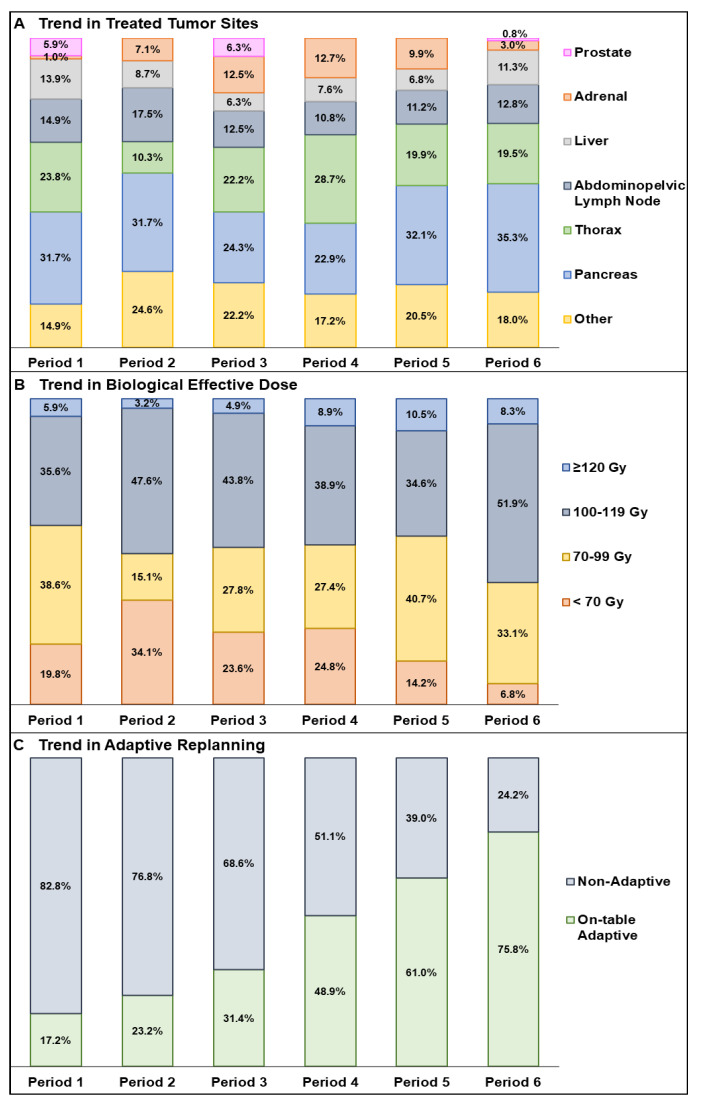
Trends in the use of magnetic resonance-guided radiotherapy by disease site (**A**), biological effective dose (**B**), and on-table adaptive vs. non-adaptive fractions (**C**) (Periods 1–6).

**Figure 4 cancers-17-00208-f004:**
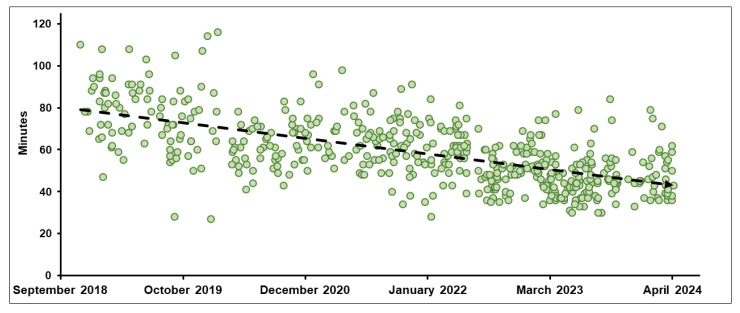
Trend in median overall total in-room time (min) for on-table adaptive radiotherapy treatment courses (Periods 1–6).

**Table 1 cancers-17-00208-t001:** Baseline characteristics of patients treated with magnetic resonance-guided radiotherapy. Abbreviations: GTV, gross tumor volume; CTV, clinical tumor volume; PTV, planning tumor volume; BED, biological effective dose.

		Tumor Sites						
Characteristics	Treatment Courses (n = 823)	Pancreas (n = 242)	Thorax (n = 172)	Abdominopelvic Lymph Node (n = 107)	Liver (n = 72)	Adrenal Glands (n = 68)	Prostate (n= 17)	Other (n = 146)
Age—yr								
Median (range)	69.0 (7.0–94.0)	71.0 (21.0–94.0)	71.0 (15.0–94.0)	67.0 (31.0–90.0)	68.0 (7.0–88.0)	64.0 (28.0–85.0)	63.5 (59.0–76.0)	69.0 (8.0–93.0)
≥65 yr—no. (%)	528 (64.2)	165 (20.0)	123 (14.9)	58 (7.0)	43 (5.2)	32 (3.9)	8 (1.0)	99 (12.0)
Sex—no. (%)								
Female	379 (46.1)	116 (14.1)	76 (9.2)	70 (8.5)	31 (3.8)	27 (3.3)	0 (0.0)	59 (7.2)
Male	444 (53.9)	126 (15.3)	96 (11.7)	37 (4.5)	41 (5.0)	41 (5.0)	16 (1.9)	87 (10.6)
Race or ethnic group—no. (%)								
Asian	13 (1.6)	7 (0.9)	3 (0.4)	2 (0.2)	0 (0.0)	0 (0.0)	0 (0.0)	1 (0.1)
Black	58 (7.0)	19 (2.3)	10 (1.2)	7 (0.9)	8 (1.0)	5 (0.6)	0 (0.0)	9 (1.1)
White	724 (88.0)	204 (24.8)	154 (18.7)	96 (11.7)	62 (7.5)	61 (7.4)	16 (1.9)	131 (15.9)
Other	10 (1.2)	3 (0.4)	3 (0.4)	2 (0.2)	0 (0.0)	0 (0.0)	0 (0.0)	2 (0.2)
Unknown/Declined	18 (2.2)	9 (1.1)	2 (0.2)	0 (0.0)	2 (0.2)	2 (0.2)	0 (0.0)	3 (0.4)
Hispanic or Latino ethnic group—no (%)								
Yes	452 (54.9)	111 (13.5)	98 (11.9)	70 (8.5)	45 (5.5)	44 (5.3)	7 (0.9)	77 (9.4)
No	371 (45.1)	131 (15.9)	74 (9.0)	37 (4.5)	27 (3.3)	24 (2.9)	9 (1.1)	69 (8.4)
Treatment Summary								
Prescribed dose—Gy								
Median (range)	50.0 (16.0–76.0)	50.0 (20.0–76.0)	50.0 (30.0–60.0)	42.0 (25.0–62.0)	50.0 (30.0–60.0)	50.0 (25.0–60.0)	40.0 (37.0–74.0)	40.0 (16.0–68.0)
Prescribed fractions								
Median (range)	5 (1–41)	5 (1–33)	5 (1–30)	5 (1–34)	5 (1–28)	5 (1–6)	5 (5–41)	6 (1–36)
Radiotherapy duration—days								
Median (range)	8 (1–67)	7 (1–64)	10 (1–65)	8 (1–54)	7 (1–39)	8 (1–27)	10 (7–59)	10 (1–67)
Prescribed BED10—Gy								
Median (range)	94.4 (28.0–200.0)	100.0 (37.5–104.6)	100.0 (36.0–149.6)	72.0 (37.5–100.0)	100.0 (48.0–200.0)	100.0 (48.0–132.0)	72.0 (64.4–87.4)	63.7 (28.0–100.8)
Fractions—no. (%)								
Total delivered	6036 (100.0)	1353 (22.4)	1460 (24.2)	631 (10.5)	396 (6.6)	307 (5.1)	91 (1.5)	1798 (29.8)
Total adapted	2643 (43.8)	1081 (17.9)	484 (8.0)	423 (7.0)	110 (1.8)	235 (3.9)	0 (0.0)	310 (5.1)
On-table adaptive courses—no. (%)	515 (62.6)	219 (26.6)	54 (6.6)	91 (11.1)	28 (3.4)	58 (7.0)	0 (0.0)	65 (7.9)
Treatment fractionation—no. (%)								
Daily	653 (79.3)	236 (28.7)	109 (13.2)	92 (11.2)	58 (7.0)	35 (4.3)	5 (0.6)	118 (14.3)
Every other day	165 (20.0)	6 (0.7)	63 (7.7)	14 (1.7)	14 (1.7)	33 (4.0)	11 (1.3)	24 (2.9)
Twice daily	3 (0.4)	0 (0.0)	0 (0.0)	1 (0.1)	0 (0.0)	0 (0.0)	0 (0.0)	2 (0.02)
Weekly	2 (0.2)	0 (0.0)	0 (0.0)	0 (0.0)	0 (0.0)	0 (0.0)	0 (0.0)	2 (0.02)
Respiratory motion management—no. (%)								
Breath-hold	687 (83.5)	235 (28.6)	160 (19.4)	69 (8.4)	69 (8.4)	67 (8.1)	1 (0.1)	86 (10.4)
Free breathing	118 (14.3)	4 (0.5)	7 (0.9)	35 (4.3)	2 (0.2)	1 (0.1)	13 (1.6)	56 (6.8)
Unknown	18 (2.2)	3 (0.4)	5 (0.6)	3 (0.4)	1 (0.1)	0 (0.0)	2 (0.2)	4 (0.5)
Abdominal compression—no. (%)								
Yes	54 (6.6)	9 (1.1)	16 (1.9)	6 (0.7)	11 (1.3)	8 (1.0)	0 (0.0)	4 (0.5)
No	769 (93.4)	233 (28.3)	156 (19.0)	101 (12.3)	61 (7.4)	60 (7.3)	16 (1.9)	142 (17.3)
Target volume—cc.								
GTV—Median (range)	18.7 (0.4–1062.0)	33.3 (1.1–854.4)	8.8 (1.0–232.4)	10.3 (0.8–1062.0)	14.5 (0.6–808.0)	15.4 (1.1–296.9)	8.7 (8.7–8.7)	284 (0.4–714.8)
CTV—Median (range)	64.9 (1.5–1450.6)	110.2 (4.7–535.4)	17.9 (2.6–410.9)	31.1 (1.6–439.0)	22.2 (1.5–129.3)	20.3 (6.5–142.3)	49.6 (27.7–68.6)	1605 (2.5–1450.6)
PTV—Median (range)	45.7 (2.4–2296.5)	63.4 (3.0–979.4)	29.9 (4.9–506.0)	25.7 (2.4–1225.2)	40.5 (4.7–618.3)	37.2 (7.7–377.4)	90.9 (34.5–124.9)	79.5 (5.1–2296.5)
Target mean dose coverage BED10—Gy								
GTV—Median (IQR)	108.2 (84.3–120.9)	113.3 (92.9–118.3)	121.2 (87.0–150.3)	93.7 (81.3–110.7)	118.6 (104.7–125.7)	113.2 (90.4–122.4)	26.5 (26.5–26.5)	69.0 (56.4–88.0)
CTV—Median (IQR)	91.4 (71.7–116.6)	93.7 (85.7–103.4)	121.1 (110.2–157.6)	87.1 (68.9–97.9)	119.9 (106.2–123.8)	98.0 (85.8–111.2)	79.0 (76.6–83.1)	62.1 (45.1–71.6)
PTV—Median (IQR)	100.1 (77.4–113.3)	106.8 (90.3–112.4)	113 (81.1–144.9)	85.7 (76.0–102.3)	112.6 (92.7–116.5)	106.3 (83.6–114.3)	76.5 (74.8–80.8)	63.4 (51.2–82.9)
Total in-room time—mins.								
Median (IQR)	51.0 (39.0–64.0)	58.0 (46.0–71.0)	45.0 (37.0–56.0)	55.0 (43.5–68.5)	49.0 (42.8–61.3)	55.5 (49.0–68.5)	45.0 (35.8–49.3)	37.5 (31.0–52.0)
Total delivery time—mins.								
Median (IQR)	15.0 (12.0–20.0)	16.0 (13.0–19.8)	17.0 (13.0–22.0)	13 (11.0–16.0)	17.5 (14.0–23.3)	17.0 (13.0–21.0)	15.0 (12.8–18.3)	12.0 (10.0–16.0)

## Data Availability

Research data are stored in an institutional repository and will be made available upon request to the corresponding authors.

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
