# Peer review of "Evolving Trends and Patterns of Utilization of Magnetic Resonance-Guided Radiotherapy at a Single Institution, 2018–2024"

_cancers, 2025, doi:10.3390/cancers17020208_

Round 1
Reviewer 1 Report
Comments and Suggestions for Authors
Thank you for the opportunity to review this manuscript. Given your vast experience with MRgRT and the variety of platforms available in your institution, I feel that the full potential of your work to help others with guiding and allocating patients to certain treatment modalities has not been fully exploited in this work.
Some specific issues to be adressed in my opinion include:
1. line 100: could you define how you decide upon the need for adaptation in your department?
2. lines 117-128: could you describe your adaptation workflow in more detail? E.g., is the recontouring limited to a certain ROI such as a PTV_expand like established by Bohoudi et al.?
3. Did you expereince any limitations regarding image quality and tracking in the two supraclavicular lymph nodes you treated?
4. lines 206-210: your rate of oART in pancreatic cases has increased to 33.6% in the most recent phase. On the other hand, overall oART rate is stated as 77.2%. Given that panceatic SBRT can be regarded as a rather challenging target volume with several radiosensitive OAR nearby, which entities/target volumes have the highest oART rates and explain the rising oART rate?
5. Does your institution have a general flow chart or workflow to allocate patients to specific treatment platforms or is the decided by individual physicians? If general recommendations or standards exist, yould you explain these?
6. line 271: Could you elaborate further on "identifying patients deriving the greatest benefit from MRgRT"? Some examples and cases would be helpful.
7. lines 339-324: Many institutions worldwide have faced challenges due to the bankruptcy of ViewRay. Could you describe the challanges you experienced and how you overcame them in more detail?
8. Given your vast experience with MRgRT, the short conclusion seems a bit unmotivated to me. Are there any more conclusions to draw from 6 years experience?
Author Response
Thank you for your review of our manuscript, “Evolving Trends and Patterns of Utilization of Magnetic Resonance-Guided Radiotherapy at a Single Institution from 2018-2024” in your journal. We have reviewed all of the comments below and provided a point-by-point response per your request.
Reviewer 1 Comments
Thank you for the opportunity to review this manuscript. Given your vast experience with MRgRT and the variety of platforms available in your institution, I feel that the full potential of your work to help others with guiding and allocating patients to certain treatment modalities has not been fully exploited in this work.
Some specific issues to be addressed in my opinion include:
Comment 1: line 100: could you define how you decide upon the need for adaptation in your department?
Response 1: We decide to adapt if the predicted dosimetry indicates that a critical OAR constraint would be violated with the original plan and/or if the target volume coverage would be improved based on the treatment day anatomy. This approach is standard across centers that offer online ART on the MRIdian Linac. Thus, our utilization of online ART was high as most of the target lesions that we treated were located in the abdomen and in proximity to GI luminal organs at risk.
Comment 2: lines 117-128: could you describe your adaptation workflow in more detail? E.g., is the recontouring limited to a certain ROI such as a PTV_expand like established by Bohoudi et al.?
Response 2: Our adaptive contouring approach is similar to many other centers. Critical OARs only within 3 cm of the PTV are contoured to ensure that the adaptive process is efficient. Contouring OARs beyond 3 cm will not influence the indication for whether online ART is needed or the quality of the adapted plan.
Comment 3: Did you experience any limitations regarding image quality and tracking in the two supraclavicular lymph nodes you treated?
Response 3: Image quality was excellent, and this is not unexpected as there is relatively little respiratory motion in this area. Respiratory motion artifact is the most common cause of poor MRI image quality.
Comment 4: lines 206-210: your rate of oART in pancreatic cases has increased to 33.6% in the most recent phase. On the other hand, overall oART rate is stated as 77.2%. Given that pancreatic SBRT can be regarded as a rather challenging target volume with several radiosensitive OAR nearby, which entities/target volumes have the highest oART rates and explain the rising oART rate?
Response 4: 77.2% represents the total number of oART courses in Periods 4-6, divided by the total number of courses treated during those periods. On the other hand, the 33.6% refers only to Period 6, comparing oART use to all other treated sites within that period. To make this clearer, we have revised the text as follows:
“When comparing the use of oART across the overall treated sites by period, the use of oART for pancreatic courses had increased from 25.0% in Period 1 to 33.6% in Period 6, while the rate for thoracic tumors had risen from 1.0% in Period 1 to 16.1% in Period 6.”
The overall incidence of oART remained high especially in the later periods as our department intentionally triaged patients to the MR-linac who would likely benefit from oART.
Comment 5: Does your institution have a general flow chart or workflow to allocate patients to specific treatment platforms or is the decided by individual physicians? If general recommendations or standards exist, would you explain these?
Response 5: Our department triages patients to the MRIdian Linac with mobile targets that would either benefit from significant dose escalation that cannot be achieved with a different treatment device using an ultra-hypofractionated treatment regimen.
Comment 6: line 271: Could you elaborate further on "identifying patients deriving the greatest benefit from MRgRT"? Some examples and cases would be helpful.
Response 6: We have expanded on the discussion of patients who may benefit from MRgRT, now referencing the case example illustrated in Figure 2 within the manuscript. To address this, we have included the following explanation:
“As shown in Figure 2, MRgRT is especially beneficial for patients with highly mobile tumors, particularly in the abdominal and pelvic regions. It allows real-time adjustments for both inter- and intra-fractional motion, improving target volume coverage and positioning to enhance dosimetric parameters. These tumors are highly sensitive to respiratory motion and are often unsuitable for MV or kV CT-guided methods, which are limited by poor soft-tissue resolution and primarily address only inter-fractional motion.”
Comment 7: lines 339-324: Many institutions worldwide have faced challenges due to the bankruptcy of ViewRay. Could you describe the challenges you experienced and how you overcame them in more detail?
Response 7: The ViewRay bankruptcy created challenges in being able to treat patients with the MRIdian Linac, primarily as device servicing was not available. Our institution made a decision to treat with the device during the bankruptcy and without service support from the company because we believed in the clinical benefit that we could provide to patients that was not available with our other advanced radiation technologies. To mitigate risk of catastrophic machine downtime we limited use of the MRIdian Linac only for patients who could not safely be treated with a similar dose on a different treatment device. We also increasingly considered single-fraction SBRT when this was feasible.
Comment 8: Given your vast experience with MRgRT, the short conclusion seems a bit unmotivated to me. Are there any more conclusions to draw from 6 years’ experience?
Response 8: We have revised the conclusion to better reflect what we’ve learned, including how our clinic has changed, which patients we have treated most frequently and the benefits of MRgRT. I appreciate your suggestion and make sure the conclusion highlights these key points. To address this, I have provided the following explanation:
“In conclusion, since the inception of our MRgRT program in 2018 we have intentionally prioritized treatment on the MRIdian Linac for patients who would benefit from significant dose escalation above what can be safely delivered on our other advanced radiation treatment systems. The unique capabilities of the MRIdian Linac enable not only safe dose escalation that frequently requires oART, especially for abdominal and pelvic tumors, but also the routine use of UHfx and increasingly single-fraction SBRT. Important reasons for our program’s success include our early decision to develop standardization of patient selection and oART workflows as well as robust training and credentialing of all staff who are involved in MRgRT within our department.”
Reviewer 2 Report
Comments and Suggestions for Authors
This article titled Evolving Trends and Patterns of Utilization of Magnetic Resonance-Guided Radiotherapy at a Single Institution from 2018-3 2024 presented very efficient research. Figures are very well prepared and are very useful to understand the workflow. I will suggest to re-do conclusion. Please consider to explain better fig 2.
- In conclusion, the final step of your paper, please remind readers of your main experimental point. Try to restate your hypothesis. In addition, please synthesize or summarize your major points and make the context of your argument clear.
Please clarify including Fig 2 in your paper and provide better explanation to the reader with few things: reason, goal and presenting results.
Author Response
Thank you for your review of our manuscript, “Evolving Trends and Patterns of Utilization of Magnetic Resonance-Guided Radiotherapy at a Single Institution from 2018-2024” in your journal. We have reviewed all of the comments below and provided a point-by-point response per your request.
Reviewer 2 comments
This article titled Evolving Trends and Patterns of Utilization of Magnetic Resonance-Guided Radiotherapy at a Single Institution from 2018-2024 presented very efficient research. Figures are very well prepared and are very useful to understand the workflow. I will suggest to re-do conclusion. Please consider to explain better fig 2.
Comment 1: In conclusion, the final step of your paper, please remind readers of your main experimental point. Try to restate your hypothesis. In addition, please synthesize or summarize your major points and make the context of your argument clear.
Response 1: Thank you for your thoughtful feedback. You’re correct, our six years of experience with MRgRT offer a chance to share more meaningful insights. I have revised the conclusion to better reflect what we’ve learned, including how our clinic has changed, which patients we have treated most frequently and the benefits of MRgRT. I appreciate your suggestion and make sure the conclusion highlights these key points. The conclusion has been modified:
“In conclusion, since the inception of our MRgRT program in 2018 we have intentionally prioritized treatment on the MRIdian Linac for patients who would benefit from significant dose escalation above what can be safely delivered on our other advanced radiation treatment systems. The unique capabilities of the MRIdian Linac enable not only safe dose escalation that frequently requires oART, especially for abdominal and pelvic tumors, but also the routine use of UHfx and increasingly single-fraction SBRT. Important reasons for our program’s success include our early decision to develop standardization of patient selection and oART workflows as well as robust training and credentialing of all staff who are involved in MRgRT within our department.”
Comment 2: Please clarify including Fig 2 in your paper and provide better explanation to the reader with few things: reason, goal and presenting results.
Response 2: We agree that the figure could better emphasize the need for oART to help readers understand the process and when adaptive replanning is necessary. To address this, we have included a more detailed explanation in the manuscript body, rather than the figure description, for clarity and conciseness.
“Figure 2 provides an example showing the importance of oART for a patient receiving an ablative dose to a mesenteric lymph node. Daily changes in the patients position of the small and large bowel relative to the GTV caused dose constraint violations, reducing the target dose coverage and requiring each fraction to be replanned.”
Round 2
Reviewer 1 Report
Comments and Suggestions for Authors
Thank you for your revisions. All my questions have been addressed and I recommend to accept the manuscript in its present form.